

# Village dogs match pet dogs in reading human facial expressions

Martina Lazzaroni[1,*], Joana Schär[2,*], Elizabeth Baxter[1], Juliette Gratalon[1], Friederike Range[1], Sarah Marshall-Pescini[1] and Rachel Dale[3]

[1] Domestication Lab, Konrad Lorenz Institute of Ethology, University of Veterinary Medicine, Vienna, Austria
[2] Department of Behavioral & Cognitive Biology, University of Vienna, Vienna, Austria
[3] Department for Psychosomatic Medicine and Psychotherapy, University for Continuing Education, Krems, Austria
[*] These authors contributed equally to this work.

Corresponding authors
Martina Lazzaroni,
martina.lazzaroni@vetmeduni.ac.at,
martina.lazzaroni@gmail.com
Rachel Dale, racheldale07@gmail.com

## ABSTRACT

Most studies on dogs' cognitive skills in understanding human communication have been conducted on pet dogs, making them a role model for the species. However, pet dogs are just a minor and particular sample of the total dog world population, which would instead be better represented by free-ranging dogs. Since free-ranging dogs are still facing the selective forces of the domestication process, they indeed represent an important study subject to investigate the effect that such a process has had on dogs' behavior and cognition. Despite only a few studies on free-ranging dogs (specifically village dogs) having been conducted so far, the results are intriguing. In fact, village dogs seem to place a high value on social contact with humans and understand some aspects of humans' communication. In this study we aimed to investigate village dogs' ability in understanding a subtle human communicative cue: human facial expressions, and compared them with pet dogs, who have already provided evidence of this social skill. We tested whether subjects were able to distinguish between neutral, happy, and angry human facial expressions in a test mimicking a potential real-life situation, where the experimenter repeatedly performed one facial expression while eating some food, and ultimately dropped it on the ground. We found evidence that village dogs, as well as pet dogs, could distinguish between subtle human communicative cues, since they performed a higher frequency of aversive gazes (looking away) in the angry condition than in the happy condition. However, we did not find other behavioral effects of the different conditions, likely due to the low intensity of the emotional expression performed. We suggest that village dogs' ability in distinguishing between human facial expressions could provide them with an advantage in surviving in a human-dominated environment.

## INTRODUCTION

Studies on dogs' cognition and behavior are quite recent and have spread since the turn of the twenty-first century when dogs started to be considered as 'real animals', worthy of investigation as domestic pets adapted to life in an urbanized environment (*Miklósi, 2014*) rather than just human accessories. Most of these studies focused on investigating

pet dogs' social skills in their interactions with humans, where subjects were shown to be extremely skillful in understanding human forms of communication. For example, pet dogs were shown to comprehend the referential nature of human pointing (*Lea & Osthaus, 2018*; *Miklósi, 2014*; *Udell, Dorey & Wynne, 2012*), follow human gaze (*Wallis et al., 2015*), social learning from humans (*Lea & Osthaus, 2018*) and overimitation (*Huber et al., 2020*), and both distinguish and appropriately respond to different human facial expressions of emotion (*Müller et al., 2015*; *Buttelmann & Tomasello, 2013*).

Dogs' socio-cognitive skills have been considered quite exceptional (*Topál et al., 2009*, but see *Lea & Osthaus, 2018*) and this raised the question of the role of domestication in affecting dog behavior and cognition (see *Range & Marshall-Pescini, 2022*; *Wynne, 2021* for a summary on the different domestication hypotheses). However, our current knowledge of dogs' socio-cognitive skills might have been widely affected by the assumption that pet dogs represent the best model for the species.

Indeed, most dog cognition studies have been performed on pet dogs living in western countries (*Aria et al., 2021*). These dogs represent a minor (around 25%, *Hughes & Macdonald, 2013*; *Lord et al., 2013*) and particular subsample of the total dog world population and contain only a fraction of dog genetic diversity (*Boyko Adam et al., 2009*). In fact, pet dogs' lives and reproduction have been entirely controlled by humans for generations, resulting in artificially maintained inbred lines (*Pilot et al., 2016*; *Pilot et al., 2015*), often selected for specific behavioral and morphological traits. Moreover, since pet dogs' lives are deeply interconnected with ours, they are highly socialized with humans, and this affects their performance in socio-cognitive tests (*Scandurra et al., 2015*). Finally, of course many pet dogs participating in cognitive tests in the labs are raised by owners who are quite sensitive to their dogs' wellness and behaviors and are often highly trained subjects, and training has been shown to affect dogs' behavior in such tests (*Marshall-Pescini, Frazzi & Valsecchi, 2016*; *Marshall-Pescini et al., 2008*; *Scandurra, Alterisio & D'Aniello, 2016*).

In reality, the vast majority of dogs (>75% *Hughes & Macdonald, 2013*; *Lord et al., 2013*) are not pets but rather are free-ranging, which constitute distinct genetic units rather than an admixture of breeds (*Boyko Adam et al., 2009*; *Pilot et al., 2015*). Since contrary to pet dogs, free-ranging dogs are free to move, to interact with other species and conspecifics and to reproduce, they are subject to both natural and sexual selection as well as selection by humans (*Pilot et al., 2015*). In fact, although some free-ranging dog populations (*i.e.,* feral dogs) living apart from humans and relying on hunting (*Krauze-Gryz & Gryz, 2014*; *Kruuk & Snell, 1981*) have been documented, the majority of free-ranging dogs usually live in close proximity to humans (*i.e.,* village dogs) and highly depend on resources provided by human activity (*Atickem, Bekele & Williams, 2010*; *Butler, Brown & Toit, 2018*; *Oppenheimer & Oppenheimer, 1975*; *Pal, 2001*). These dogs are still facing the selective forces of the domestication process and are the most globally ubiquitous type of dog, and thus should be considered the best study model to investigate the effect of domestication on dogs' behavior and cognition (*Range & Marshall-Pescini, 2022*).

Despite the fact that free-ranging dogs should represent an important study subject, only few behavioral studies on social interactions between free-ranging dogs (specifically village dogs) and humans have been conducted so far. Overall, results from these studies
are consistent and suggest that village dogs place a high value on social contact with humans and might have some abilities in understanding human forms of communication (*Bhattacharjee et al., 2020*; *Bhattacharjee, Sau & Bhadra, 2018*; *Bhattacharjee & Bhadra, 2022*; *Brubaker et al., 2019*; *Lazzaroni et al., 2020*). These skills could provide subjects with an advantage in surviving in a human dominated environment. For example, village dogs have shown sensitivity to human attentional states; subjects looked significantly longer to an attentive than an inattentive human which provided them with food. Interestingly, pet dogs and shelter dogs tested with the same paradigm did not differ in the duration of looking at the experimenter when he/she was attentive or inattentive, likely due to different motivation towards humans determined by previous lifetime experiences and/or genetic factors (*Brubaker et al., 2019*). Further support of village dogs' ability to read human communicative cues has also been provided by another study which found that village dogs were able to distinguish between a neutral, a friendly (experimenter bending forward and extending both the arms) and a threatening (experimenter raising his hand with or without a wooden stick) human, when given the possibility of obtaining food or social contact. The results from this study suggest that village dogs can use human social cues as indicators of the potential risks and benefits of approaching (*Bhattacharjee, Sau & Bhadra, 2018*). Finally, village dogs were also capable of reading more complex human cues such as different types of pointing in order to locate hidden food items (*Bhattacharjee et al., 2020*; *Bhattacharjee & Bhadra, 2022*). The observation of such skills in free-ranging dogs, which are largely acquired through experience (*Wynne, 2021*), might suggest their possible adaptive role.

With the current study we aimed to further investigate village dogs' ability in understanding human communication, specifically their ability to distinguish between different human facial expressions. Previous studies found evidence that domesticated animals including cats, horses, goats and pet dogs can differentiate between human facial expressions and react appropriately to the valence of the expressions (*Nagasawa et al., 2011*; *Müller et al., 2015*; *Albuquerque et al., 2016*; *Albuquerque et al., 2018*; *Galvan & Vonk, 2016*; *Smith et al., 2016*; *Nawroth et al., 2018*; *Quaranta et al., 2020*). For example, pet dogs could discriminate between neutral and smiling faces (*Nagasawa et al., 2011*) and looked significantly longer at the face whose expression was congruent to the valence of a vocalization (*Albuquerque et al., 2016*). Pet dogs also learned to discriminate between happy and angry human faces that were presented only partially (upper or lower half of the face) and to transfer their training contingency to novel faces expressing similar emotions (*Müller et al., 2015*). Finally, pet dogs performed significantly more mouth-licking (a putative stress/appeasement signal, *Pedretti et al., 2022*) when looking at an angry human face than at a smiling face (*Albuquerque et al., 2018*) and show varying tail wagging behaviors according to the type of emotional stimulus presented (*Quaranta, Siniscalchi & Vallortigara, 2007*).

However, since emotions are expressed differently in different species, discriminating emotional expressions in heterospecifics might be particularly challenging and the ability to recognize such expressions might be highly dependent on experience (*Barber et al., 2016*; *Müller et al., 2015*). Thus, while being able to understand human facial expressions could

be adaptive for village dogs, giving them important information relating to the risk-benefit of approaching (or leaving) a human, it is an open question whether their relatively limited experience (compared to pet dogs) is sufficient to develop such a competence.

We tested and compared adult village dogs and pet dogs in their ability to distinguish between neutral, happy, and angry human facial expressions in a test mimicking a potential real-life situation. In the test an experimenter was sitting on a chair and repeatedly performed one of three facial expressions (happy, angry, neutral) while eating a piece of food. The person then "accidentally" dropped the food on the ground, just in front of her feet. The experimenter did not make any vocalizations or gestures. Thus, the dogs had to choose whether to approach the person in close proximity in order to obtain the food based on the facial expression alone.

In line with previous studies conducted on pet dogs we hypothesized that animals from this population would be able to distinguish between the three facial expressions and behave accordingly. However, in line with the increasing evidence of village dogs' abilities in understanding human social cues and the potentially high adaptive value this ability may have for village dogs, we could also hypothesize that they too would be capable of distinguishing between facial expressions. Nevertheless, considering pet dogs' more frequent exposure to human faces in their daily life, it can be predicted that pet dogs will be better at distinguishing facial expressions than village dogs.

More specifically, if subjects were able to distinguish between different human facial expressions and attribute the proper valence to them, we predicted that they would:

-P1_spend more time in proximity to the experimenter in the happy condition than in the neutral and angry conditions, as it is an overtly positive signal from the human, as opposed to the aversive (angry) or uncertain (neutral) natures of the other expressions.

-P2_be more likely to eat all available food in the happy than in the neutral and angry conditions, since approaching the human in close proximity will be less 'risky' in the happy condition.

-P3_wag their tail for longer in the angry condition than in the happy and neutral conditions, since tail wagging is an affiliative/submissive behavior performed by dogs (and wolves) as a post-conflict management strategy, which can function to reduce aggression (*Cools, Van Hout & Nelissen, 2008*; *Lazzaroni, Marshall-Pescini & Cafazzo, 2017*; *Walters et al., 2020*).

-P4_perform more gaze aversion in the angry than the happy and neutral conditions, since gaze aversion is considered a submissive, stress or appeasement behavior (*Cools, Van Hout & Nelissen, 2008*; *Firnkes et al., 2017*; *Pedretti et al., 2022*; *Srithunyarat et al., 2018*).

-P5_look longer at the experimenter in the happy and neutral conditions than in the angry condition (*Somppi et al., 2016*).

## MATERIALS & METHODS

### Ethical statement

Ethical approval for this study was obtained from 'Ethik und Tierschutzkommission' of the University of Veterinary Medicine (Protocol number: ETK-08/09/2018 and ETK-16/01/2019). Informed consent was obtained by all owners of the pet dogs prior to testing.

The authorization to test the village dogs was provided by the municipality of Taghazout (Morocco).

## Subjects and study area
### Village dogs (Vd)

The study area for the village dogs was focused on Taghazout in Morocco, in an area of around 4 km$^2$ including the town of Taghazout and the surrounding beaches and villages. Taghazout is a touristic town of around 5000 inhabitants, characterized by the presence of tourist facilities such as restaurants, markets, and shops. Around two million dogs live in Morocco, most of them are freely moving and can be considered as free-ranging dogs. The population used in this study lives in the municipality of Taghazout and can be considered 'village dogs' (*Coppinger & Coppinger, 2001*). Despite having regular interactions with humans, village dogs, including those in the current sample, do not have owners who provide them with medical treatments or confine them in houses, controlling their activities. Village dogs are born on the streets and beaches in human proximity, where food sources are easily reachable (*Sen Majumder et al., 2016*), and are free to move and reproduce (*Pal, 2011*). Preliminary genetic analyses showed that our tested population is genetically akin to free-ranging dog populations from Eastern Europe and not representative of breed dogs (*Pilot et al., 2015*; M Pilot, 2018, personal communication). The village dogs of the current sample feed by scavenging on human garbage produced by the local people or touristic activities (restaurants, hotels *etc.*). Besides, locals and tourists feed dogs directly, sometimes on a regular basis at specific locations. Taghazout is known to have a positive attitude to, and high tolerance of, dogs, indeed during regular observations 75–90% of observed human-dog interactions in the study area were positive (R Dale, 2018, personal communication).

In this study, a total of 72 village dogs were tested (42 females; 30 males). Three female experimenters (JG, LB, RD) travelled by car to look for village dogs in their natural environment in the municipality of Taghazout. Particular attention was paid to choosing dogs that were solitary to avoid interference by conspecifics and on including only adult dogs (appearing to be over 1 year of age because of their size and behavior). All dogs were unfamiliar to the experimenters.

### Pet dogs (Pd)

Pet dogs were tested at private gardens or at dog-areas in public parks in the city of Vienna and at the Wolf Science Center in Ernstbrunn, Austria; a few dogs were additionally tested in private gardens in Baselland, Switzerland. All pet dogs tested were unfamiliar to the experimenter. Two groups of pet dogs were tested in different environments. The first group of pet dogs (PdA), 63 dogs of various breeds (32 females; 31 males) were tested in 4 outdoor areas in dog parks in Vienna (areas' size ranging from 869 m$^2$ to 0.1 km$^2$) or in a large testing enclosure at the Wolf Science Center (950 m$^2$). People and other dogs could be present in the dog areas or in the surroundings. Female experimenters (JS, ML, RD) recruited the dogs by approaching owners in outdoor areas and asking if they wanted to participate in the test with their dog.

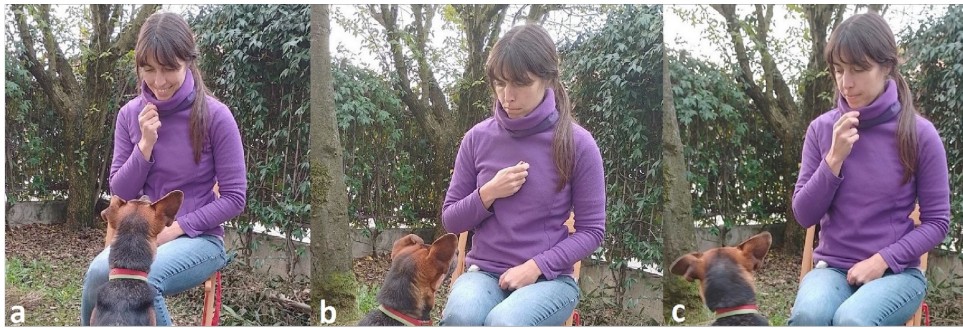

**Figure 1 Example of the three facial expressions.** Examples of the experimenter expressing a happy (A), angry (B) and neutral (C) facial expression.

We tested village dogs (Vd) in what would be considered their normal, natural outdoor habitat. It can be argued that going to a dog park, although regular, may be more exciting and distracting for pet dogs than the home streets are for Vd (*Lazzaroni et al., 2020*). Therefore, to control for this factor, to optimize the experimental setting and to control for distractions from the surroundings, we additionally tested a group of dogs in their home outdoor environment (garden, PdG). Two female experimenters (JS, RD) tested a total of 53 pet dogs (33 females; 20 males; age: adult dogs only (>1-year-old)). The dogs were recruited by an online announcement and by directly contacting them over the database of dog owners of the Clever Dog Lab (University of Veterinary Medicine of Vienna). In this case, the owners were told to not feed their dogs in the two hours before the test.

## Test procedure

The testing procedure was slightly different for the pet dogs and village dogs since it had to be adjusted to the different environments. All dogs were tested only once in one of the three possible conditions, the test condition was randomly assigned to the subjects, with the condition that each expression was presented the same number of times within each group (See Fig. 1 for facial expressions, Fig. 2 for testing procedure, and Video S1 for examples of tests with village dogs in the three conditions: https://figshare.com/articles/media/Video1_avi/21747449). Videos were checked by authors who could identify the individual dogs of the study area to ensure no dog was tested more than once.

### Village dogs (Vd)

The experimenters located solitary dogs while moving around the village with a car. Once a suitable dog was located, one helper got out of the car and approached the dog to get its attention. Meanwhile the experimenter set up the testing scene, which contained a chair and a tripod mounted with a video camera within a two-meter distance. Even though the aim was to have as few outside influences as possible, it was not possible to test the dogs in a completely distraction-free area. If the dog's attention was on the experimenter during the eating and facial expression phases, it was considered as a success. Two helpers were responsible for recording the experiment with a camera and distracting other dogs or

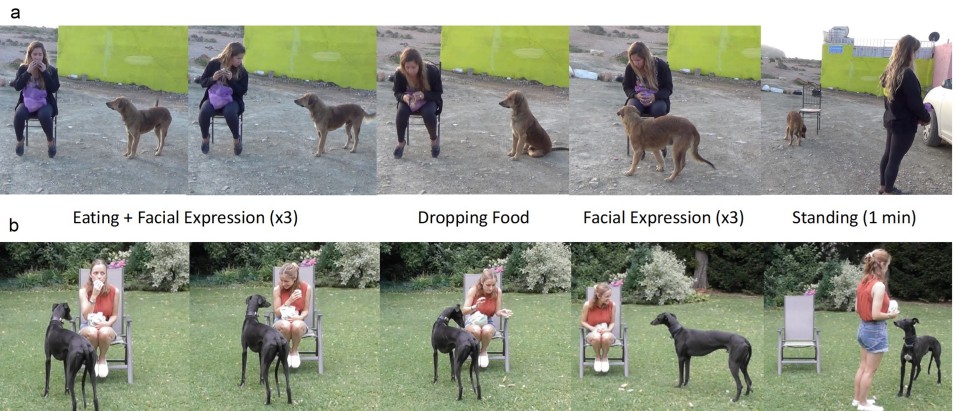

| | | | |
|---|---|---|---|
| Eating + Facial Expression (x3) | Dropping Food | Facial Expression (x3) | Standing (1 min) |

**Figure 2** **Test set up and procedure with village dogs (A) and pet dogs (B).** In both cases, the experimenter sat on a chair and ate sausages and biscuits. Then she paused eating and made eye contact while making one of three facial expressions (happy, angry, neutral). The procedure of eating (for 5 s), which was followed by making the facial expression (for 5 s), was repeated three times. After that, the experimenter dropped the food and then alternated again between looking away from the dog and making the facial expression three times (x3). Then she stood up and faced away from the set-up for one minute. After the standing time, the experiment ended.

humans that might disturb the course of the experiment, while the experimenter ran the trial.

The starting position included the experimenter sitting on the chair and holding one piece of sausage with one piece of cracker. On her lap there was a bag of extra sausages and crackers. The helper who interacted with the subject dog from the very beginning would now bring the subject over to the set-up and then stand motionless behind the video camera at 2m distance. The trial began when the dog was focused on the experimenter. The experimenter ate the cracker for approximately 5 s (by counting internally), looking down at the food. The experimenter ate the food to capture the interest and increase the motivation of the subjects. Then she looked at the dog and once she made eye contact with the dog, she expressed for approximately 5 s one of three facial expressions: happy, angry, or neutral (Fig. 1). The remaining time the experimenter kept a neutral face and looked away from the dog. The happy facial expression included a smile, the angry expression included a frown, and the neutral would exhibit a straight mouth with no other movement. After the facial expression, the experimenter dropped the gaze and ate again. These alternating phases between eating and showing a facial expression were repeated three times, to potentially increase the clarity of the facial expression for the subjects and ensure they had paid attention to it. Then, the experimenter ate the food one more time and "accidentally" dropped it on the ground. If the dog looked away or got too distracted during the experiment, the experimenter would cough or rustle the bag in her lap to regain its attention, but never spoke. The experimenter only dropped the food when she was sure that the dog was attentive to her. After dropping the food, the experimenter repeated three times the same facial expression as before each lasting approximately 5 s. The experimenter started to perform the facial expression only when the dog was looking at her. This action

was performed regardless of whether the dog had already approached to eat the food on the ground or not. After the third and final facial expression, the experimenter stood up, walked about 1-2 m away from the dog in between the chair and camera, and faced away from the set-up for one minute. During the standing time, the experimenter did not make any movements and ignored the dog if it approached. The trial ended after this phase. LB tested 27 dogs, RD tested 25 dogs and JG tested 20 dogs.

### Pet dogs (Pd)

The first group of pet dogs (PdA) was tested in outdoor areas in Austria. Three female experimenters (JS, ML, RD) were involved. The owner of the pet dog was present during the experiment, and they sat on a second chair 2-3 m to the side of the experimenter (facing away from the experimental set up) reading a newspaper, ignoring the dog no matter what it did throughout the whole experiment. The experimenter ignored the dog while explaining the experiment to the owner and while setting up the experiment. The trial began when the dog was focused on the experimenter. The remaining process of the experiment was the same as described for the village dogs. JS tested 39 dogs, ML and RD each tested 12 dogs.

The second group of pet dogs (PdG) were tested with the same procedure, but this time in the owner's garden with fewer distractions. The owners were asked to remain indoors for the duration of the trial, unless for example a dog was very dependent on the owner's presence in order to be calm and focused or there was no possibility for the owner to go elsewhere, in which case they sat on a chair facing away, as described above. Two experimenters (JS, RD) were involved in the experiments; JS tested 43 dogs and RD tested 10 dogs.

Each session was videotaped by a video camera (Panasonic HD-Camcorder HC-W580). The three conditions performed were counterbalanced across the experimenters.

## Statistical analyses

All videos were coded using Solomon coder (developed by András Péter, Dept. of Ethology, Budapest, solomon.andraspeter.com). The trial started when the experimenter began eating while sitting on the chair and it ended after 60 s of standing time of the experimenter. See Table 1 for definitions of the coded behaviors. We additionally recorded subjects' sex and, to control for the possible effect of subject's motivation for food in the willingness of approaching the experimenter, all dogs were classified into two body condition groups. We used the scale of the *WSAVA Nutritional Assessment Guidelines Task Force Members (2011)* under ideal (thin dogs) and ideal/over ideal (normal to fat dogs) body condition. Among pet dogs, no subjects resulted in having an under ideal body condition, while we classified 16 village dogs as under ideal.

The videos from the village dogs were coded by LB and JG. Inter-observer reliability was carried out between three experimenters (LB, JG, ML), each coding the same 10 videos out of 72 videos (Intra-class correlation coefficient, two-way model using the irr package: Eating available food ICC = 1; Gaze aversion ICC = 0.8; Proximity ICC = 0.97; Looking at experimenter ICC = 0.86; Tail wagging ICC = 0.97). The videos from the pet dogs

**Table 1   Description of the coded behaviours.**

| Behaviour | Description |
| --- | --- |
| Eating available food (frequency) | The subject eats the all the available sausage and the biscuit. |
| Gaze aversion (frequency) | The subject has eye contact with the experimenter and then looks away, turning its head to one side. |
| Proximity (duration[a]) | The subject is within one dog body length distance of the experimenter. |
| Looking at the experimenter (duration[a]) | The subject's head is oriented towards the experimenter's face. |
| Tail wagging (duration[a]) | The subject wags the tail from side to side. |

Notes.
[a]Behaviour durations were coded in seconds.

(PdA and PdG) were coded by JS. Inter-observer reliability was carried out between two experimenters (JS, ML), each coding the same 20 videos out of 117 videos (Intra-class correlation coefficient: Proximity ICC = 0.974; Eating available food ICC = 0.806; Tail wagging ICC = 0.849; Gaze aversion ICC = 0.826; Looking at experimenter ICC = 0.876).

Statistical analyses were run in R (version 4.0.2, *R Core Team, 2021* using Generalized Linear Mixed Models (GLMM, *Baayen, 2008*) (see Table S7). After fitting the models, we inspected the distribution of the individual specific deviations from the common intercept and slopes (BLUPs). Collinearity of predictors, assessed by applying the function vif of the R package car (*Fox et al., 2012*) appeared not to be an issue (*Quinn & Keough, 2002*) (Table S6). Overdispersion appeared not to be an issue (range of dispersion parameters 0.936–1.124). We determined model stability by dropping levels of the random effects one at a time and comparing the estimates derived from models fitted on the respective subsets with those obtained for the full data set (see Tables S1–S5). We obtained confidence intervals of model estimates by means of a parametric bootstrap ($N = 1,000$ bootstraps; function simulate of the package glmmTMB or bootMer of the package lme4) (see Tables S1–S5). For all models, to keep type I error rate at the nominal level of 5% (*Barr et al., 2013*; *Schielzeth & Forstmeier, 2009*) we included all theoretically identifiable random slopes components (condition, sex and body condition within experimenter ID). *P*-values for the individual effects were based on likelihood ratio tests comparing the full model with the respective reduced models lacking the model predictors (R function 'anova') (*Barr, 2013*).

### Proximity, Tail wagging and Looking at experimenter models (predictions P1, P3, P5)

In models 'Proximity', 'Tail wagging' and 'Looking at experimenter' we tested whether the proportion of time (sec) individuals spent in proximity with the experimenter, or tail wagged at the experimenter, or looked at the experimenter differed between groups and between conditions. To this end, we ran three GLMMs with beta error distribution and logit link function (*Bolker, 2008*; *McCullagh & Nelder, 1989*), using the function glmmTMB of the equally named package (version 1.0.0 *Brooks et al., 2017*) (Proximity model, Tail wagging model and Looking at experimenter model) including group (PdA, PdG, Vd), condition (happy, angry, neutral) and their interaction as fixed effects. We also added sex (male or female) and body condition (thin or normal) into the models to control for their effects and included the identity of the experimenter (experimenter ID) as a random factor. To test for differences between groups we compared each one of the full models described

above with a null model lacking group, condition, and their interaction, but otherwise identical to the full model. Since this comparison resulted in significance we further tested the significance of the interaction by comparing the full model with an identical reduced model lacking the interaction between group and condition. This comparison did not result in significance, thus we removed the interaction from the model and inspected the output. This procedure was applied to all three models.

### Eating available food model (prediction P2)

We tested whether the probability of eating all available food or not (1/0 response) differed between groups and between conditions. To this end, we ran a GLMM with a binomial distribution using the function glmer of the package lme4 (*Bates et al., 2015*), including group (PdA, PdG, Vd), condition (happy, angry, neutral) and their interaction as fixed effects, the identity of the experimenter as random factor. We also added in the model sex and body condition to control for their effect. We compared the full model with a null model lacking the predictors group, condition, and their interaction but otherwise identical to the full model. Since this comparison resulted in significance, we tested the significance of the interaction by comparing the full model with a reduced model lacking the interaction between group and condition. We tested *post-hoc* comparisons using the function emmeans of the equally named package (version 1.7.0).

### Gaze aversion model (prediction P4)

We tested whether the frequency of gaze aversions differed between groups and between conditions. To this end, we ran a GLMM with a negative Poisson distribution using the function glmer.nb (*Bates et al., 2015*), including group (PdA, PdG, Vd), condition (happy, angry, neutral) and their interaction as fixed effects, the identity of the experimenter as random factor. We included the test duration (log transformed) as an offset term. We also added in the model sex and body condition to control for their effect. To test for differences between groups we compared the full model described above with a null model lacking group, condition, and their interaction, but otherwise identical to the full model. Since this comparison resulted in significance, we further tested the significance of the interaction by comparing the full model with an identical reduced model lacking the interaction between group and condition. This comparison did not result in significance, thus we removed the interaction from the model and inspected the output.

## RESULTS

### Proximity (P1)

We tested whether the duration of staying in proximity to the experimenter depends on the dog group and/or the displayed facial expression. We found that pet dogs in gardens and in outdoor areas remained in proximity to the experimenter for longer time periods than village dogs (PdA-Vd: $z = 5.770$, $p < 0.001$; PdG-Vd: $z = 5.808$, $p < 0.001$), irrespective of the facial expressions displayed (interaction between group and condition was not significant, full-reduced model comparison: $\chi 2 = 5.722$, $df = 4$, $p = 0.221$) (Fig. 3A). We found no differences in the time spent in proximity to the experimenter between the two

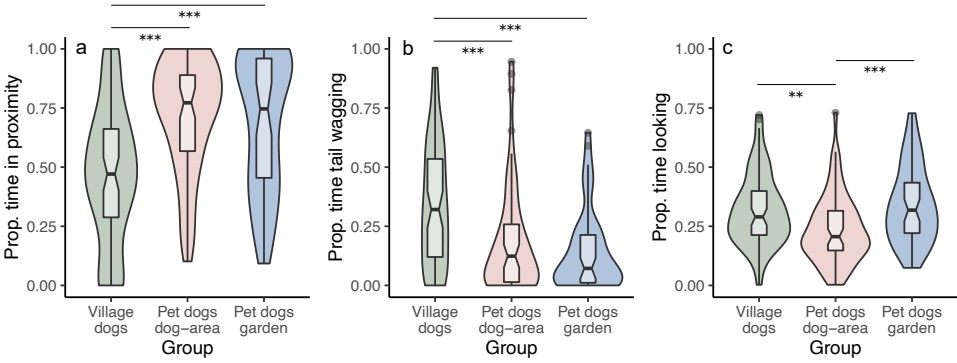

**Figure 3** **Proportion of time spent in proximity, tail wagging and looking at the experimenter.** Violin plots with inlaid boxplots demonstrating median, first and third quartile and minimum and maximum of the proportion of time that subjects spent in (A) proximity with the experimenter, (B) tail wagging at the experimenter, (B) looking at the experimenter for village dogs (Vd, $N = 72$), pet dogs tested in outdoor areas (PdA, $N = 64$) and pet dogs tested in gardens (PdG, $N = 53$). The violin plot outlines the kernel probability density with the width of the shaded area indicating the proportion of data at each location. $p$-value $\leq 0.01 = **$; $p$-value $\leq 0.001 = ***$.

groups of pet dogs ($z = 0.341$, $p = 0.733$). Notably, within the village dogs, those with a body condition classified as "thin" stayed in proximity with the experimenter for longer than subjects classified as having a "normal" body condition ($z = 2.425$, $p = 0.015$).

We did not find differences between facial conditions in the time spent in proximity to the experimenter (*i.e.,* no main effect of condition; angry-happy: $z = -0.784$, $p = 0.433$; angry-neutral: $z = -0.971$, $p = 0.332$; happy-neutral: $z = -0.490$, $p = 0.624$). See Tables S1A and S1B.

### *Eating available food (P2)*
We tested whether the probability to eat all the available food (dropped by the experimenter) was dependent on the dog group and/or the displayed facial expression. Overall, the full model was clearly significant as compared to the null model (full-null model comparison: $\chi 2 = 36.735$, $df = 8$, $p < 0.0001$) and revealed a significant interaction (interaction between group and condition: $\chi 2 = 11.95$, $df = 4$, $p = 0.018$). The *post-hoc* comparison revealed differences between village dogs and pet dogs, specifically with pet dogs tested in dog areas showing a significantly higher probability of finishing all food in comparison with village dogs in the happy and neutral conditions, but not in the angry condition (*post-hoc* comparison: happy condition $z = 3.314$, $p = 0.01$; neutral condition $z = 3.983$, $p < 0.001$). See Table S2.

### *Tail wagging (P3)*
We tested whether the duration of tail wagging depends on the dog group and/or the displayed facial expression. We found that village dogs wagged their tails at the experimenter for longer than both groups of pet dogs (PdA-Vd: $z = -3.302$, $p < 0.001$; PdG-Vd: $z = -3.896$, $p < 0.001$), irrespective of the facial condition displayed (interaction between group and condition was not significant, full-reduced model comparison: $\chi 2 = 6.133$,

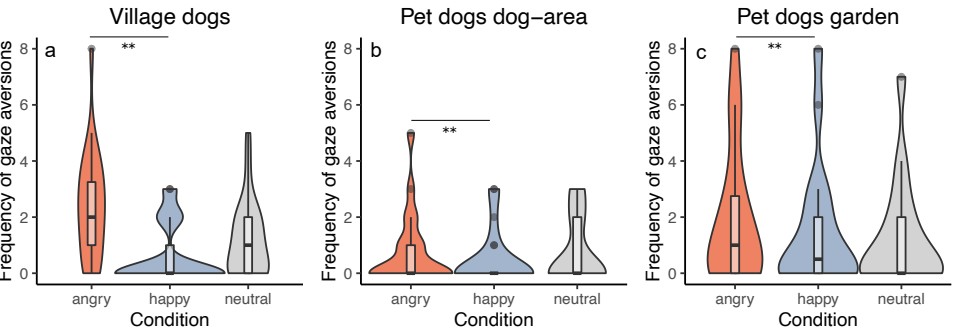

**Figure 4 Frequency of gaze aversion.** Violin plots with inlaid boxplots demonstrating median, first and third quartile of frequency of gaze aversion during the experiment for (A) village dogs (Vd, $N = 72$), (B) pet dogs tested in outdoor areas (PdA, $N = 64$) and (C) pet dogs tested in gardens (PdG, $N = 53$) and in dependency of the facial expression displayed (angry, happy, and neutral facial expression). The violin plot outlines the kernel probability density whereby the width of the shaded area indicates the proportion of data for each location. $p$-value $\leq 0.01 = $ **.

$df = 4$, $p = 0.189$) (Fig. 3B). We found no differences in the duration of tail wagging between the two groups of pet dogs ($z = -0.765$, $p = 0.444$), and no differences in tail wagging were observed between facial conditions in all groups (angry-happy: $z = -0.351$, $p = 0.725$; angry-neutral: $z = 0.049$, $p = 0.960$; happy-neutral: $z = -0.397$, $p = 0.691$). See Tables S3A and S3B.

### Gaze aversion (P4)

We tested whether the frequency of performing gaze aversions by dogs depends on the dog group and/or the displayed facial expression. All groups responded similarly to the facial expressions (interaction between group and condition was not significant, full-reduced model comparison: $\chi 2 = 5.500$, $df = 4$, $p = 0.240$), averting gaze more frequently in the angry than in the happy condition ($z = -2.956$, $p = 0.003$) and showing a tendency to more frequent gaze aversion in the angry than in the neutral condition ($z = -1.909$, $p = 0.056$). We did not find differences between the neutral and the happy condition ($z = 1.081$, $p = 0.279$). Generally, we also found that pet dogs tested in private gardens (PdG) and village dogs (Vd) performed more gaze aversions than pet dogs tested in dog areas (PdA) (PdG-PdA: $z = 3.269$, $p = 0.001$; Vd-PdA: $z = -4.404$, $p < 0.001$), whereas no difference was apparent between pet dogs tested in gardens and village dogs ($z = -1.185$, $p = 0.236$) (Fig. 4). See Tables S4A and S4B.

### Looking at experimenter (P5)

We tested whether the duration of looking at the experimenter (the dog's head is oriented towards the experimenter's face) depends on the dog group and/or the displayed facial expression. We found that, overall, pet dogs tested in gardens (PdG) and village dogs (Vd) looked at the experimenter for longer than pet dogs tested in dog areas (PdA) (PdG-PdA: $z = 3.837$, $p < 0.001$; Vd-PdA: $z = -2.813$, $p = 0.005$; Vd-PdG: $z = 1.115$, $p = 0.265$), irrespective of the facial expression displayed (interaction between group and condition was not significant, full-reduced model comparison: $\chi 2 = 5.580$, $df = 4$, $p = 0.233$)

(Fig. 3C). There was also no main effect of condition on the duration of looking at the experimenter (angry-happy: $z = -0.915$, $p = 0.360$; angry-neutral: $z = -0.482$, $p = 0.630$; happy-neutral: $z = 0.351$, $p = 0.725$). See Tables S5A and S5B.

## DISCUSSION

With the current study we aimed to investigate village dogs' ability to read human facial expressions by comparing them with pet dogs. Overall, our results suggested that despite their lower socialization experience with humans, village dogs might be able to distinguish between subtle human communicative cues, suggesting that this competence may be relevant for dogs' survival in their natural environment.

We found that both village dogs and pet dogs behaved similarly, performing more gaze aversions when the experimenter was expressing an angry face than a happy one. We additionally found a tendency for all dogs to perform more gaze aversion during the angry facial expression than the neutral one (confirming prediction P4). Averting gaze has been considered either a submissive intraspecific behavior in dogs (*Bonanni et al., 2017*; *Cools, Van Hout & Nelissen, 2008*), or a reaction to an opponent's aggressive behavior in an agonistic context (*Van der Borg et al., 2015*). A recent experimental study found evidence that averting gaze represents a displacement behavior elicited by high levels of arousal. In fact, the authors found a positive relationship between the overall frequency of aversive gazes and the levels of cortisol in pet dogs and a higher frequency of aversive gazes in dogs facing a frustrating situation than a positive anticipation situation (*Pedretti et al., 2022*). Indeed, pet dogs' arousal has been previously found to be affected by different human facial expressions, with dogs watching the image of an angry face having higher heart rate (*Barber et al., 2016*) and enlarged pupil sizes (*Karl et al., 2020*; *Somppi et al., 2017*) than dogs watching a happy face. Thus, we suggest that in our study both pet dogs and village dogs were able to distinguish between the different human facial expressions, since they appeared more stressed in the angry condition than in the happy (and neutral) condition as demonstrated by the higher number of aversive gazes.

Although the previous result highlights pet dogs' and village dogs' ability in reading human facial expressions, contrary to our predictions we did not find other effects of condition on the dogs' behaviors. In fact, there were no differences between conditions for all dog groups in the time spent in proximity with the experimenter (P1), the probability of eating all the food that was thrown on the ground (P2), the duration of tail wagging (P3) or looking at the experimenter (P5). The absence of some of these differences might be because the angry condition, which solely consisted of a threatening face, was not enough to elicit in dogs an active avoidance of the experimenter (*i.e.,* close proximity, eating the food and looking at the experimenter) who was instead strongly attractive because she was holding food. Evidence that the proximity with the experimenter might have been affected by subjects' motivation for food is supported by the fact that, independently from the test condition, village dogs that were classified as "thin" and which might have had a high motivation for food, spent more time in proximity with the experimenter than village dogs classified as having a "normal" body condition. Future studies could indeed include
control conditions where the emotional expressions are performed without the presence of food.

Moreover, although dogs appear quite skillful in reading human facial expressions in an artificial experimental setting (*Albuquerque et al., 2018*; *Albuquerque et al., 2016*; *Müller et al., 2015*; *Nagasawa et al., 2011*), it is possible that in a real-world situation, the facial expression alone is not enough to affect subjects' approach/proximity behavior (*Correia-Caeiro, Guo & Mills, 2023*). In fact, in such a naturalistic context the absence of body language and/or vocalizations might be confusing for the dogs, who have been found to rely on bodily emotional expressions to interpret humans' emotions (*Correia-Caeiro, Guo & Mills, 2021*). Indeed, other studies which included human body language other than facial expression, and mimicked a possible real-life situation, observed a congruent reaction of dogs (both pet dogs and village dogs) towards different humans' emotional expressions (*Bhattacharjee, Sau & Bhadra, 2018*; *Merola et al., 2014*; *Vas et al., 2005*). Therefore, gaze aversion may reflect a more subtle measure of facial expression discrimination, at least for negative expressions.

It is interesting to notice village dogs' ability in reading subtle human communicative cues, such as facial expressions, despite the common assumption that they spend relatively little time in interacting with humans (*Majumder, Chatterjee & Bhadra, 2014*). Although recent studies highlighted that some populations of village dogs do indeed interact intensively with humans (*Bhattacharjee & Bhadra, 2020*), no study ever directly investigated differences in the extent of social interaction with humans between populations of pet dogs and village dogs. Our findings are not surprising since previous studies already highlighted village dogs' skillfulness in reading other aspects of human communication (*Bhattacharjee et al., 2020*; *Bhattacharjee, Sau & Bhadra, 2018*; *Bhattacharjee & Bhadra, 2022*; *Brubaker et al., 2019*). However, it is remarkable that the likely different extent of social experience with humans of village dogs and pet dogs did not seem to have determined major differences between the two populations. Indeed, dogs' life experience has been previously found to affect subjects' performance when it comes to reading human communication. For example, studies investigating dogs' skillfulness in human-guided tasks found that adult pet dogs outperformed young pet dogs (*Bray et al., 2021*; *Dorey, Udell & Wynne, 2010*; *Zaine, Domeniconi & Wynne, 2015*), pet dogs outperformed shelter or kennel dogs (*D'Aniello et al., 2017*; *Jarvis & Hall, 2020*; *Lazarowski & Dorman, 2015*; *Udell, Dorey & Wynne, 2010*; *Zaine, Domeniconi & Wynne, 2015*), and trained pet dogs outperformed untrained pet dogs (*McKinley & Sambrook, 2000*). Since the ability to read human communication does not impact pet dogs and shelter dogs' survival, it might solely depend on the amount of their exposure to humans. On the contrary, this is not the case for village dogs, where subjects that are better in reading human cues might have higher chances of survival in a human dominated environment. Thus, a limited exposure with humans might be enough for them to develop complex social skills.

Interestingly, although life experience did not seem to differently affect pet dogs' and village dogs' ability in discriminating human facial expressions, it caused some overall differences between the two populations. In fact, independently from the condition, village dogs spent less time in close proximity with the experimenter than pet dogs, and tail

wagged for longer. Recent evidence suggests that rather than being the consequence of an increase in subjects' emotional arousal (*Beerda et al., 1999*; *Travain et al., 2016*), tail wagging represents a communicative signal (*Pedretti et al., 2022*). Tail wagging is used as an intraspecific post-conflict management strategy that reduces aggression between former opponents (*Cools, Van Hout & Nelissen, 2008*; *Lazzaroni, Marshall-Pescini & Cafazzo, 2017*; *Walters et al., 2020*) and is considered a possible signal of formal submission in dog-human interaction (*Döring et al., 2014*; *Gácsi et al., 2005*; *Rehn & Keeling, 2011*; *Yong & Ruffman, 2014*). Additionally, we observed that 94% of pet dogs tested in dog areas, 87% of pet dogs in gardens, but only 41% of village dogs, started to eat the food as soon as it was dropped, before the experimenter made the final three facial expression demonstrations. Thus, village dogs might have been more cautious when interacting with the experimenter compared to pet dogs, and this is supported both by the shorter time spent in proximity with the experimenter and the longer tail wagging. These differences might be either due to village dogs' poorer socialization experience with humans, which makes them less confident than pet dogs and/or to their more frequent experience of negative interactions with humans (*Paul et al., 2016*). Despite the evidence that social interaction with humans might result in a negative outcome, it appears clear that benefits could overcame the risks, especially for subjects that can read human communication (*Bhattacharjee, Sau & Bhadra, 2018*).

Finally, a proper comparison between dog populations differing in their life experience would require testing all subjects in the same test conditions. Otherwise, it would not be possible to disentangle the role of test setting and life experience in affecting potential differences on subject's behaviors. However, village dogs can only be tested unleashed in their home environment, which is usually full of distractions, while pet dogs are usually tested in a restricted environment, often indoors, and in the presence of their owner. To partially account for (and to investigate) the possible effect of the test setting on dogs' behaviors, we included in our study two groups of pet dogs tested in different settings (dog areas in public parks and private gardens). As previously observed (*Brubaker et al., 2017*; *Lazzaroni et al., 2020*), also in this study we found that overall pet dogs tested in dog areas seemed quite distracted, since they spent less time looking at the experimenter and consequently performed fewer aversive gazes than pet dogs tested in gardens and village dogs. However, other than the test setting, such differences might have been due to the fact that pet dogs tested in dog areas were recruited randomly in the parks while pet dogs tested in private gardens were recruited from a list of dogs who usually participate in cognitive tests and are typically highly trained subjects. The evidence that pet dogs' behavior might vary greatly depending on test settings and past training history should highlight the importance of testing populations of free-ranging dogs, whose behaviors are congruent to the selective forces that act in their natural environment. Including "wild" populations in experimental studies, such as free-ranging dogs, would indeed provide a better understanding of the function of different behaviors and cognitive skills, other than the context in which such abilities evolved (*Rosati, Machanda & Slocombe, 2022*).

Overall, all dog groups showed more gaze aversion in the angry condition, but no other behavioral effects. This suggests that facial expressions alone may not be enough to counter

the high motivation for food in the likelihood of displaying active approach behaviors. However, subtle measures such as gaze aversion suggest an ability for village dogs, as well as pet dogs, to discriminate facial expressions and should be explored further.

## CONCLUSIONS

We compared village dogs and pet dogs in a socio-cognitive task investigating their ability in reading different human facial expressions. Both village dogs and pet dogs seemed to be able to distinguish between facial expressions since they performed a higher frequency of aversive gazes in the angry condition than in the happy condition. These results are in line with the most recent findings on village dogs' socio-cognitive skills, supporting their ability in reading human communication. We further suggest that, compared to pet dogs, whose behavior might be highly affected by breed, past training history and widely different life experiences, free-ranging dogs represent a better study sample to investigate the effect of the domestication process on dog behavior and cognition.

## ACKNOWLEDGEMENTS

We thank the owners of the pet dogs that participated in this study. We thank the authorities and the population of Taghazout (Agadir, Morocco) for allowing us to run the tests with the free-ranging dogs and providing useful information regarding the dogs. Finally, we thank the association 'Stray Dogs International Project' that introduced us to Taghazout.

### Funding

This work was supported by the DOC fellowship of the Austrian Academy of Sciences and the Austrian Science Fund (FWF) project P34675-G. The funders had no role in study design, data collection and analysis, decision to publish, or preparation of the manuscript.

### Grant Disclosures

The following grant information was disclosed by the authors:
DOC fellowship of the Austrian Academy of Sciences and the Austrian Science Fund (FWF): P34675-G.

### Competing Interests

The authors declare there are no competing interests.

### Author Contributions

- Martina Lazzaroni conceived and designed the experiments, performed the experiments, analyzed the data, prepared figures and/or tables, authored or reviewed drafts of the article, and approved the final draft.
- Joana Schär performed the experiments, analyzed the data, prepared figures and/or tables, authored or reviewed drafts of the article, and approved the final draft.

- Elizabeth Baxter performed the experiments, authored or reviewed drafts of the article, and approved the final draft.
- Juliette Gratalon performed the experiments, authored or reviewed drafts of the article, and approved the final draft.
- Friederike Range conceived and designed the experiments, authored or reviewed drafts of the article, and approved the final draft.
- Sarah Marshall-Pescini conceived and designed the experiments, authored or reviewed drafts of the article, and approved the final draft.
- Rachel Dale conceived and designed the experiments, performed the experiments, authored or reviewed drafts of the article, and approved the final draft.

**Animal Ethics**

The following information was supplied relating to ethical approvals (*i.e.*, approving body and any reference numbers):

Ethik und Tierschutzkommission of the University of Veterinary Medicine (Protocol number: ETK-08/09/2018 and ETK-16/01/2019).

**Field Study Permissions**

The following information was supplied relating to field study approvals (*i.e.*, approving body and any reference numbers):

The authorization to test the village dogs was provided by the municipality of Taghazout (Morocco).

**Data Availability**

The output of the behavioral coding from videos (raw data), and additional information regarding statistics, are available in the Supplementary Files.

The example video is available at figshare: Lazzaroni, Martina (2023). Video1.avi. figshare. Media. https://figshare.com/articles/media/Video1_avi/21747449.

**Supplemental Information**

Supplemental information for this article can be found online at http://dx.doi.org/10.7717/peerj.15601#supplemental-information.

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
