# Peer review of "Village dogs match pet dogs in reading human facial expressions"

_PeerJ, doi:10.7717/peerj.15601_

## Round 0.1 · original submission · Major Revisions

Decision on PeerJ80290

First, I must apologize for the delay in getting all of the reviews in for your manuscript. I think you will agree that the reviewers had very helpful suggestions for your paper and it was worth the wait for their comments. All of the reviewers find the topic to be of value and the research to be generally well conducted. However, they do have questions about the methods and statistical approach which require clarification. Reviewer 1 also calls for the need to more clearly define and operationalize your constructs. I agree with the reviewers’ assessments and I have some additional comments of my own.

What is meant by “Since free-ranging dogs are still facing the natural selective forces of the domestication process..” in the abstract? I saw what you meant by line 71-72 and I completely agree that this is an important point given that domestically bred animals are not subject to sexual selection. I would make sure it is clear that you mean the population of free ranging dogs (if they were produced in the ‘wild’) rather than individuals. I would be concerned that this is a messy distinction if free ranging dogs sometimes breed with dogs that were bred by humans and then abandoned, for example. I think you should be more explicit about how these “village dogs” fit into the large picture. Do we know that they were all free-ranging from birth? How do they compare to wild non-domestic dogs? What do they have in common with wild dogs and companion domestic dogs? What about other non-domestic species that nonetheless interact frequently with humans? I think more care should be given to the study sample and what your expectations would be based on their experiences with humans and the traits selected for through the process of domestication.
I agree that more details are needed to understand the procedure. If the experimenter performed the facial expression three times for 5 sec each after dropping the food, what expression were they presenting in between? Shouldn’t they have held that expression the entire time? Or been neutral at least in between?

It is not clear if each dog was exposed to only a single expression. That is, was the emotion expression presented between-subjects? This should be absolutely clear and it should be indicated how many dogs received each condition.

Why did you not use latency to approach as an outcome?

The use of violin plots is unusual. Can you justify why you selected this mode of presentation?

It would have made sense to pilot this study with one group of dogs – the pet dogs - before going to the effort of testing all three groups. Given that there are very slight effects of condition, even though you would expect it for the pet dogs, you should have modified the procedure to make the emotions more salient – perhaps include vocal or postural cues as well. There is no reason to assume that pet dogs are not attuned to human expressions so it seems it would be useful to develop a procedure that shows these effects before comparing the ability of village dogs. Now it is a bit difficult to say anything about the results other than that the methods may have been insufficient to produce the expected effects. So perhaps you should focus the discussion more on the overall differences between the dog groups in their approach behavior without overstating their ability to differentiate the emotions.

I think the conclusion is a bit overstated on line 433-435 since you found only subtle effects of gaze aversion and eating. The gaze aversion finding isn’t surprising since direct eye contact is threatening for many species in interspecific interactions.

What do you mean by “real animals” on line 45?

Be sure to insert commas between clauses.

You might also cite Buttelmann, D., & Tomasello, M. (2013). Can domestic dogs (Canis familiaris) use referential emotional expressions to locate hidden food?. Animal cognition, 16, 137-145.

You should cite other studies that have used very similar methodology to test the responses of, for example, domestic cats, to humans exhibiting different expressions. I believe there are papers doing something similar in horses, pigs and other species as well.

How can you be sure you did not re-test any dogs more than once?

·

Basic reporting

The study is well-structured and clearly described. The English language seems to be professional throughout (not being a native speaker, I might overlook some minor errors, if there are). Maybe the authors can give a statement if a proficient English speaker has checked the paper.

The introduction gives a sufficient background on the investigated topic citing adequate literature as far as I can judge, it is a little bit long, though. I have some suggestions to improve clarity. I will mention the main topic of each paragraph so that the authors can better follow my ideas.

Paragraph 1 is on the socio-cognitive skills of dogs. Being not an expert in this particular theme, I wonder whether the list mentioned in lines 49-51 is the complete list of socio-cognitive skills when interacting with humans and whether socio-cognitive skills are a synonym for “understanding human forms of communication”. The authors mention human pointing, human gaze, and human facial expressions of emotions. In summary, it would be helpful to have a better concept of “socio-cognitive skills”.

Paragraph 2 is on pet versus free-ranging dogs as study model. I have only two suggestions here:

(1) Line 70, phrase “In reality, the vast majority of dogs (75% Hughes and Macdonald, 2013; Lord et al., 2013) are not pets but rather are free-ranging.” I would add a “>” before the percentage as Lord et al. 2013 talk of 17-24% restricted dogs.

(2) In line 75, the authors give a definition for village dogs, i.e., dogs living in close proximity to humans mentioning four references. I think that it is necessary (maybe not necessarily in the introduction, but in the methods part) to further define what kind of village dogs the authors refer to in their study. For example, it is not clear whether those dogs do have one or several “owners” or “care-takers” or not at all (i.e., eating only garbage).

Paragraph 3 refers to the findings of earlier studies with regard to socio-cognitive skills of free-ranging dogs. Paragraph 4 mentions one single study on the recognition of communicative cues in free-ranging dogs, while Paragraph 5 is on understanding pointing in free-ranging dogs. I think that those three paragraphs should be integrated into one paragraph only, while I wonder whether paragraph 5 might be shortened or some parts moved into the discussion, as this paper is not on pointing, but on the recognition of human facial expressions.

Paragraph 6 introduces to the study aims and then refers to previous literature on the ability of pet dogs to differentiate human facial expressions. Paragraph 7 then refers to the knowledge gap, i.e., that it is unknown whether village dogs acquire the competence to understand human expressions despite their lacking experience with humans. I think that the study aims should be presented in a clearer way, not separated by a paragraph on previous knowledge on pet dogs´ facial recognition. Maybe that paragraph could be placed next to the new paragraph 3 where the authors report on studies of free-ranging dogs on the recognition of human cues. Afterwards, they can explain the knowledge gap and what study has been performed.

Additionally, in line 111, is human social cues used as a synonym for human social communication? If there are several terms on these concepts (as stated earlier, socio-cognitive skills etc.), I suggest the authors only use one for consistency as not all possible readers might be perfectly aware of the nuances of these concepts.

Paragraph 8 presents the experimental design and hypothesis. Although I think this is a bit detailed for the introduction (I first thought this paragraph should be completely part of the methods), maybe the authors need such detail for explaining their predictions. It is ok for me, although I found that part longer than I am used to from other studies.

Paragraph 9 presents the predictions. Here I strongly recommend to use a nicer formatting, I would use P1-P5 to name their predictions and then refer to those predictions throughout the text and particularly in the discussion.

With regard to the figures, they illustrate the main findings and are easy to understand as they are kept consistent. I wonder though, why Figure 3, 4 and 6 are not presented in the same way as Figure 5, i.e., they could be summarized into one figure showing the three graphs in one. This helps improving the consistency (i.e., size of figures, same arrangement) of figures and understanding as the reader can better compare the three graphs at once. Figure 5 and the new summarized figure should have letters for each subgraph, so that the authors can refer to that letter in the figure legend. Although I understand the abbreviations Vd, PdA and PdG, those are not intuitive. Isn´t it possible to either change the abbreviations (or not use them) or add icons above them, e.g., a dog with a collar for Vd a dog without a collar with several trees standing for parks for PdA and the same dog with flowers standing for a garden in PdG. This is only a suggestion. Futher, I could not find that the authors mention the unit of time in Figures 3, 4, and 6 (seconds). With regard to Figures 1 and 2, A and B in the legend should be a and b or majuscules be used in the figure itself.

Raw data is supplied. The excel of the raw data should anonymize the dog names. As the authors mention that they use a list of dog owners frequently participating in studies, the names of their dogs might be familiar to other investigators, hence, anonymity is not guaranteed as should be expected from studies using ethical statements.

Experimental design

The research is original primary research within the scope of the journal. The research question is very well defined and highly relevant as free-ranging dogs are deficiently studied as the authors state in the introduction.

In the Materials & Methods section, under ethical statement and section Subjects and study area, Village dogs, I wonder whether the village dogs studied in Morocco did not have owners or caretakers to ask permission for. Otherwise, all ethical issues seem fine and back-upped by supplementary material.

When selecting the village dogs, did the experimenters aspire a balanced selection with regard to the sex of the dogs? That dogs “appeared to be over 1 year of age” was decided based on the size and behavior of the dogs?

Test procedure
Can the authors please explain what exactly was the adjustment of the procedure to the different environments? Looking on the photographs of Figure 2, is it only about different types of chairs (which would be a minor thing)? Please clarify.

Line 201. When mentioning the two figures and the video (I could not find that video among the materials provided) I would suggest to give a short explanation what the reader can expect from the figure, i.e. (see Fig. 1 for facial expressions, Fig. 2 for village versus pet dog tests, Video 1 for xy). In Fig. 3 (x3) probably means three times, please explain this abbreviation in the figure legend.

I wonder why the experimenter ate food. Can this be better explained, please? Probably you presented this stimulus as you expected a higher level of interest in the dog, being more willing to stay near the experimenter and enter into contact, but this is not mentioned neither is literature cited.

Statistical analyses
Lines 287-308, the authors describe the modelling approach in general first, and in specific for the different models later. The problem is that in the general part, random effects are mentioned, but not specified, among others (for example, model validation is already mentioned, which is supposed to be done at the end of modelling, but then, in the single models (lines 310, 323, 331) information earlier mentioned is repeated and modelling steps prior to model validation are mentioned. In short terms, I find the structure confusing. To make this clearer, I would first describe the general approach used (i.e., Generalized Mixed Models), then explain the data exploration valid for all models, i.e., overdispersion etc. Afterwards explain the three model approaches mentioning the error distributions, packages used etc., together with a table that gives an overview of the model approaches: model name, response variable, explanatory variable, random factors. Then you can finish with a general paragraph on how you assessed model validity. I strongly recommend that overview table.

Some more doubts on statistics. I understand that modelling has a variety of approaches and maybe the authors have used an approach I am not familiar with, but I wonder whether the missing of some frequent information on modelling can be provided.
(1) Did you not use the Akaike Information Criterion for model selection? Or is this approach not necessary as you have only one explanatory variable (condition?) and several random factors?
(2) Can the interaction term be better explained?
(3) What exactly means moderate stability?
(4) Line 337 You stated “We first tested the significance of the interaction by comparing the full model with an identical reduced model lacking the interaction between group and condition and since this comparison did not result significant, we compared the full model with a null model lacking the predictors group, condition, and their interaction.” I do not understand this condition, only when the interaction is not significant, the authors compare full and null models. I understand that full and null models should always be compared when proceeding to model validation. Secondly, is it not normal to try all parameter combinations? I am really a bit lost here.
(6) Please provide the VIF value. Did you also run pair correlations between the parameters?

Otherwise, all methods are very well described and with sufficient detail to replicate the experiment.

Validity of the findings

The results section is clearly written and understandable, but I have doubts whether the statistics are correctly and sufficiently reported.

Normally, the results of modelling are given as estimates and/or Akaike weights (e.g., Grimm et al. 2015, doi: 10.1002/ece3.1824). Here the authors report z values which I do not understand well. Also, where can I find the results of model validation (i.e., bootstrapping)? Did the authors also inspect residual plots?

The discussion is well written. I have some suggestions to restructure, though: Instead of first (second paragraph, line 436 ff) giving the result of gaze aversion with angry faces, I would start with whether the five predictions were fulfilled or not (line 453-464), only then would I discuss the second most important finding about gaze aversion (which was similar in both groups of dogs, but this was not the aim of the study). Now, lines 436-452 can be stated and then you can continue discussing the experimental setting (body language) you started in line 465. Otherwise, you first talk about gaze aversion, then about not fulfilled predictions, and again about gave aversion.

Line 452: Does a new paragraph start here? I can´t see this from formatting, but I suggest to start a new one.

Line 454/455: Please name the behaviors in the order previously reported.

Line 464: The authors could suggest how future experiments should be designed if the holding of food by the experimenter was a possible bias.

Line 478: I wonder whether it is really true that village dogs interact less with humans. Maybe it would be interesting to measure this variable in future studies and/or evaluate whether it matters that pet dogs interact more with the same persons and village dogs probably more with diverse persons, but during less time. The unknown background of village dogs might be some sort of problem in these kinds of studies. Maybe this point merits some further thoughts.

Line 484: Please insert a comma after “For example”.

Line 529: The authors suggest to include wild populations in experimental studies, what exactly do they refer to? Feral dogs? Dingoes? How would it be possible to include them in experimental situations? Maybe this thought must be better explained.

Conclusions
I think the conclusions are fine, but I wonder whether the authors can be more specific and strengthen what they state in lines 523-528, i.e., that conclusions drawn from single populations of dogs might be biased from the dog background, experience, and test situation.

Line 541: Please write socio-cognitive, not socio cognitive.

Additional comments

I find the topic really very interesting and it is a novel research line to include free-ranging, not pet dogs, in those types of studies. I strongly recommend the publication of an improved version.

Reviewer 2 ·

Basic reporting

This is an interesting piece of work comparing pet and village dogs' abilities to differentiate human facial expressions. The authors highlighted the influence of socialization on pet dogs' interactions with humans but correctly emphasized the adaptive value of the same for village dogs. Such comparative studies are timely, and essential to understand the domestication of dogs and largely the evolution of socio-cognitive traits.

I enjoyed reading the introduction and especially the discussion of the manuscript. The discussion is thoroughly written with relevant information and conclusive to what they actually found. However, my main concern is the methods section which lacks clarity (See later for specific points).

The authors provided raw data, however, the codes were not given. I would like to see the same.

Experimental design

The predictions seem to be convoluted. Moreover, the experimental procedures need more clarity. Please see below my specific comments -

Introduction:
Line 148-150 - What made authors predict that the dogs will eat food only in happy as compared to neutral and angry conditions? Whereas, in most other predictions, happy and neutral conditions were clubbed together?
Line 151 - Tail wagging was introduced out of nowhere in one of the predictions. It was only present in the discussion. Please at least mention in one or two sentences in the introduction.
Line 151-159 - Gazing from distance or in close proximity? Tail wagging and gazing together from distance or in close proximity?

Methods:
Line 170 - Please provide an overview of the habitat where village dogs were tested, i.e., approx. area size, and relevant habitat details, e.g., presence of people, shops, etc.
Line 173-174 - "A lot of time" sounds vague. Did you quantify how much time? or could you cite any previous work to provide an estimation?
Line 176 - "..known to have a positive attitude." - What is the basis for this statement? Did you conduct surveys? or objectively looked at the dog-human interactions?
Line 183 - Please rephrase assuming the entire city of Vienna can't be the study area. Use something like 'pet dogs were tested at...'
Line 185-188 - This section is important to visualize the study yet crucial details are missing. Please provide area size and potential distractors, which may influence dogs' behavior.
Line 193-195 - Please check sentence structure.
Line 217 - How far was the camera placed?
Line 219 - How was this 5 second eye contact achieved? Both experimenter and subject looked at each other for 5 seconds? Please clarify.
Line 223 - Please explain why the alternating phases were repeated three time? Was there a specific reason or this was done to ensure that the expressions are sufficient enough to induce a response.
Line 227-229 - Didn't the dogs eat the food rewards before commencement of all three trials? If so, how many times you have seen that?
Line 230-233 - Then what does the second phase tell us?
Line 250-253 - How did authors ensure that the owners have no influence (even if sitting and ignoring the subject) on the subject dogs?
Line 282 - Please specific which ICC type.
Line 292 - Please use "negative binomial' instead of "negative Poisson" throughout the manuscript.
Line 294 - We do not have any information about the number of food pieces, whether they were identical for every condition, etc. Varying number of food pieces can affect the related durations significantly. Therefore, the probability might not be the best representation here. Why not 1/0 response?
Line 320 - Not clear which was the full model? with or without interaction terms? Also, see line 338.

Validity of the findings

Line 392 - I wonder why authors chose to use frequency of gaze aversion and not duration. Was the sole reason to do so because duration of gazing and duration of gaze aversion are mutually exclusive? (provided eating is not factored in). While frequency is still a good measure, duration of gaze aversion, in my opinion, would have been more appropriate as it would clearly indicate dogs' understanding of the different conditions (therefore, not limited to just subtle behavioral indications!)

Line 462-646 - Please discuss this finding.
Line 476-478 - Although the authors highlighted that village dogs spend little time interacting with humans, please refer to Bhattacharjee and Bhadra 2020, which showed humans a robust contributor to dogs' social interaction networks.

Additional comments

I am curious to know why authors chose the term "village dogs" over "free-ranging dogs" or "free-roaming dogs". Does this mean that these dogs are only found in villages and not in cities? Since this study did not quantify habitat type, i.e., city or rural/village, I would like to see an explanation.

·

Basic reporting

This manuscript reports the results of an extensive study carried out to compare the ability of village dogs in Africa and pet dogs (tested in Vienna) to read human facial expressions. The authors report that both sets of dogs can distinguish between angry and happy and/or neutral facial expressions, and this is revealed through higher frequency of gaze aversions in case of angry faces, as compared to the others, by all dogs. The article is well written and gives extensive background information relevant to the study.

Experimental design

The experiment design is simple and suitable for answering the question. The sample sizes are robust and data analysis has been quite rigorous.

Validity of the findings

I have only one comment regarding the results and their interpretation. The conclusion drawn by the authors is based on the rate of gaze aversion, but the other important result that dogs do not avoid spending time around humans showing the angry expression, and do not shy away from taking the food dropped by them has been ignored in the interpretation of the results. I think this is important, and should be discussed at length.

Based on the results, one can indeed say that the dogs show different responses to the various human facial expressions, but whether they actually distinguish between these expressions to a level to affect their interaction with the human is not clear, due to the presence of food in the experiment. As the authors have suggested, the motivation might have been too high due to the presence of food. Ideally, a control condition, with the human facial expressions intact, and no food being eaten or held by the experimenter. This I find to be the only weakness in this study.

Additional comments

I think mentioning the lack of a control condition is important. The conclusions should be toned down a little. I appreciate that the authors have highlighted the importance of studying free-ranging dogs to understand dog behaviour.

---

## Round 0.2 · Minor Revisions

Both reviewers are happy with the revisions that you have made to your manuscript - they have enhanced the research communication of the work you have carried out; so that is fantastic! Just some minor issues to attend to in regard to the second reviewer.

Reviewer 2 ·

Basic reporting

Thank you for addressing the concerns and fixing the issues.

Experimental design

N/A

Validity of the findings

N/A

·

Basic reporting

I am happy to read the revised version of this manuscript. I see that the authors have substantially revised the manuscript, primarily by providing further details on the experimental procedure and analysis. This definitely improves the manuscript. As I have already mentioned before, this is an interesting piece of work that deserves publication. The literature review is quite extensive and the text is written in a clear language.

Experimental design

I would like to ask for a minor clarification. Please see below.

Validity of the findings

No further comments.

Additional comments

I would like to request the authors to provide one clarification. From the video, the procedure of the experiment is mostly clear. However, it would be good if they can clarify how the experimenter kept tab of the 5s duration for showing an expression to the dog.

---

## Round 0.3 · accepted · Accept

I am happy with the changes made according to the minor issues raised by the reviewer, and now happy to accept your paper in PeerJ. I hope you are able to advertise your research widely to attract interest and engagement.